# Modeling Energy Expenditure Estimation in Occupational Context by Actigraphy: A Multi Regression Mixed-Effects Model

**DOI:** 10.3390/ijerph181910419

**Published:** 2021-10-03

**Authors:** André Lucena, Joana Guedes, Mário Vaz, Luiz Silva, Denisse Bustos, Erivaldo Souza

**Affiliations:** 1Department of Engineering and Environmental Sciences, Engineering Centre, Federal Rural University of Semi-arid Region, Mossoró 59625-900, RN, Brazil; andrelucena@ufersa.edu.br; 2Associated Laboratory for Energy, Transports and Aeronautics, LAETA (PROA), Faculty of Engineering, University of Porto, 4200-465 Porto, Portugal; gmavaz@fe.up.pt (M.V.); ldbs@fe.up.pt (D.B.); 3Department of Production Engineering, Federal University of Paraíba, João Pessoa 58051-970, PB, Brazil; bueno@ct.ufpb.br (L.S.); elopesouza@gmail.com (E.S.)

**Keywords:** energy expenditure, metabolic assessment, mixed-effects model, occupational health

## Abstract

The accurate prediction of energy requirements for healthy individuals has many useful applications. The occupational perspective has also been proven to be of great utility for improving workers’ ergonomics, safety, and health. This work proposes a statistical regression model based on actigraphy and personal characteristics to estimate energy expenditure and cross-validate the results with reference standardized methods. The model was developed by hierarchical mixed-effects regression modeling based on the multitask protocol data. Measurements combined actigraphy, indirect calorimetry, and other personal and lifestyle information from healthy individuals (*n* = 50) within the age of 29.8 ± 5 years old. Results showed a significant influence of the variables related to movements, heart rate and anthropometric variables of body composition for energy expenditure estimation. Overall, the proposed model showed good agreement with energy expenditure measured by indirect calorimetry and evidenced a better performance than the methods presented in the international guidelines for metabolic rate assessment proving to be a reliable alternative to normative guidelines. Furthermore, a statistically significant relationship was found between daily activity and energy expenditure, which raised the possibility of further studies including other variables, namely those related to the subject’s lifestyle.

## 1. Introduction

The accurate prediction of energy requirements for healthy individuals has many useful applications [1]. Various studies associated with energy expenditure have been conducted within different contexts. From the occupational perspective, it has also been proven to be of great utility for improving workers’ ergonomics, safety, and health [2,3,4,5,6,7,8,9].

The most accurate measurement methods considered are the double-labeled water method and direct and indirect calorimetry by oxygen consumption rate [2,10]. Alternatively, as direct measurements are complex, expensive, and unsuitable for field studies, several models have been adopted as a significant technique for assessing energy requirements [11]. Most of them, developed by regression methods, combine anthropometrics, physiological variables, and movement patterns and are accepted for predicting energy requirements over a wide variety of activities [2,12,13].

Concerning applicable current standards, ISO 8996:2004 [10] is the most relevant and sophisticated one, providing four different levels of accuracy for estimating or determining the metabolic rate, spanning from basic methods (occupation and activity classification) to the previously referred expertise procedures (double-labeled water and direct calorimetry). However, most methods involve some limitations for field measurements and are based on reference values only applicable to specific populations [14].

Furthermore, other alternative methods have emerged based on physical activities quantification to estimate energy expenditure, providing practical advantages. One of these methods is actigraphy, a technique based on movement monitoring with specific criteria to define sleep and wake states, offering minimal subject burden, versatility, and relative cost-efficiency [15,16]. The actigraphs are composed of multidirectional accelerometers, but some have other sensors such as gyroscopes, inclinometers, GPS, and light sensors. Studies on energy expenditure by actigraphy have been developed in both research and clinical settings, to analyze sleep–wake patterns [17], to examine the metabolic demands associated with specific occupational environments [18,19], and to evaluate energy requirements among patient groups [20] and resulting from free-living activities [21].

However, despite their advantages, when using accelerometers, researchers face a challenge identifying which of the available regression equations is best to predict energy expenditure for their study case [22,23]. There are indeed several equations to estimate energy expenditure using actigraphy. Most of these were developed by regression methods using oximetry, some physiological variables, and movement variables [15]. These current regression techniques collect and average accelerometer counts over a specified period and then are used to estimate energy expenditure following the relationship dictated by the prediction equation [15].

More recently, and despite not being widely adopted for use, machine learning modeling of raw accelerometer data has also emerged as a method with strong potential for improving estimates of physical activity and energy requirements from accelerometers [24]. Artificial neural networks, for example, have been applied to estimate energy expenditure by using, as input, demographic variables, physiological variables, and accelerometers signal features that vary during physical activities [24,25]. However, there is no clear evidence of the physiological variables influence on the outcomes of these calculation models, the most efficient number of variables, or the prioritization recommendations of specific variables to be inserted in the models.

In the face of the various methods to estimate energy expenditure, current research trends involve applying one or more prediction techniques and a small number of variables. On the other hand, from the occupational perspective, energy expenditure estimation is associated with work-related health and wellness, evaluating thermal stress and workload, and preventing musculoskeletal disorders and physiological stress [26]. Currently, there is no comprehensive research in which various methods are compared, encompassing an extensive range of characteristics to determine energy requirements from various activities applicable to occupational settings [26].

For occupational contexts, field measurement challenges are found that make the double-labeled water and indirect oximetry unfeasible to apply due to environmental issues, cost, and time restrictions [27]. In addition, tasks variability, multibody movements, and different levels of physical intensity combined make the assessment using a single model a challenging goal to achieve [12,13]. With the increased number of tasks with light or sedentary activities, it is difficult to generalize the energy expenditure for a diverse workforce, supporting the increased importance of personal factors (sex, age, and body composition, among others) in the assessment approach [28]. As a result, aiming to surpass some of these challenges [29,30], methods based on actigraphy combined with other calculation techniques give relevant and accurate results where the application of reference methods is unpracticable [31].

Regarding calculation methods, studies show that there is a diversity of techniques used to calculate energy expenditure [12,13], ranging from log-linear regression [24,32], support vector machine [12,33,34], decision trees [35], Bayesian classification [30], and regression networks [34]. Artificial neural networks are found in various configurations, such as the radial basis function network—RBFN, the generalized regression neural network—GRNN [36], and the multilayer perceptron feedforward artificial neural network—MLP-ANN [37].

Other alternatives involve modeling using hierarchical mixed regression or multilevel regression. Among various features, the multilevel regression technique allows incorporating the hierarchical nature of the data into the analyses, integrating variables measured at different levels of the hierarchy, and examining how regression relationships vary across clusters [38]. Using these modeling techniques makes it possible to decompose the variance and identify its portions: the explained variance that can be attributed by variation in such level and the unexplained variance (or residual variance) [39].

This work aims to propose an actigraphy-based model for energy expenditure estimation in the occupational context based on hierarchical regression modeling, and cross-validate the results obtained with the gold standard (indirect calorimetry) and other validated methodologies (normative guidelines from ISO 8996:2004 and predictive equations). The work is directly guided to occupational enforcement without restricting its application to other contexts covered by the proposed protocol and model.

## 2. Materials and Methods

The experimental trials were conducted at the Laboratory on Prevention of Occupational and Environmental Risks (PROA) at the Faculty of Engineering of the University of Porto and the Faculty facilities. The volunteers were fully informed of the details of the experimental procedures and were briefed on the purpose, potential risks, and benefits of the experiences. Written consent was read and signed by them prior to starting the trials. The Ethics Committee of the University of Porto approved the study (Report 106/CEUP/2021).

### 2.1. Sampling

Volunteers between the ages of 18 and 40 years, who declared themselves to be nonsmokers, not frequent users of supplements or dietary supplements, not to have a medical history or any other health condition that would indicate risks or compromise metabolic or cardiorespiratory functions, and not to have physical limitations or other restrictions of movement, were invited to participate in the experiment. The volunteers were invited to participate in the research by formal e-mail messages, and by disclosure on social networks.

Participants were excluded if they had metabolic diseases, cardiovascular diseases or limiting conditions, kidney diseases, diabetes, hypertension, neoplasms, pregnancy or suspected pregnancy, exertional asthma, allergies to any element or activity present in the experiment, condition or alteration of limbs that interferes with body movements, obesity above level I, depression, alcoholism, or suffering from the use of other substances.

### 2.2. Participants

A total of 54 subjects fulfilled the inclusion criteria and were tested. Four were excluded for not completing all the activities defined in the experimental protocol (Section 2.4).

As a result, 50 healthy men and women (16% black, 42% white, 40% mixed race and 2% Asian) between the ages of 18 and 40 years were included in the study and their physical characteristics are shown in Table 1. In general, participants were considered mentally and physically healthy, were nonsmokers and did not report any disease or medication known to alter metabolic rate nor physical or movement restrictions. Of them, 32% were considered to have a sedentary lifestyle, 42% were moderately active, and 26% were highly active.

Before the experience, each participant completed the informed written consent and two questionnaires. The first questionnaire involved general personal information, eating habits, clinical history, and physical activity level. Questions related to the level of physical activity were based on the Portuguese version of the International Physical Activity Questionnaire (IPAQ) [40]. The second questionnaire addressed inquiries related to sleeping time and quality and stress, anxiety, and depression indicators. Information on sleeping quality and time was collected using the short version of the Pittsburgh Sleep Quality Index questionnaire (PSQI) [41]. Similarly, anxiety, depression, and stress experienced at that moment were assessed through the 21-question version of the Depression Anxiety Stress Scales (DASS) [42].

### 2.3. Instrumentation and Facilities

The experiments were mostly conducted in a climatic chamber (FITOCLIMA 25000EC20). This chamber (3.20 m × 3.20 m) simulates an office environment and controls temperature to an accuracy of ±0.2 °C and relative humidity ±5%.

#### 2.3.1. Indirect Calorimetry

Energy expenditure was measured from pulmonary gas exchange using a breath-by-breath portable gas analyzer (Cosmed K4b2, Rome, Italy).

#### 2.3.2. Accelerometer-Derived Activity

For recording activity patterns, participants were asked to wear three accelerometers wGT3X-BT (ActiGraph, Pensacola, FL, USA) on their dominant limb in the wrist and ankle and on the right side of the hip. These devices are small in size and lightweight; 3.3 cm × 4.6 cm × 1.5 cm and 42.6 g, respectively. They are sensitive to accelerations from 0.05–2.0 G’s and have a band-limited frequency of 0.25–2.5 Hz. The ActiGraphs were set to sample at a rate of 30 Hz, and an 8-bit A/D converter digitized the signal. They were initialized to collect data in 1 s epochs, and the results were downloaded directly to a compatible computer using a USB cable. The cutpoints for free life activities were defined for the magnitude vector according to Freedson et al. [43].

#### 2.3.3. Physiological Monitoring

A Polar H7 heart rate sensor (Polar Electro, New York, NY, USA) was used to measure cardiac activity, synchronizing its data with the recordings from the actigraphs and the K4b2. An eq02+ Equivital monitor chest belt (Hidalgo, Cambridge, UK) was used for physiological monitoring to record heart rate, respiratory frequency, skin temperature, and movement by accelerometry. As a result, 1.8 kg were added to account for the additional weight of the devices.

Figure 1 exemplifies the equipment’s disposition, with the portable metabolic analyzer in the anterior part of the body (portable unit, battery, mask), an ambient thermometer in the shoulder, and the actigraphs in the wrist, waist, and ankle. According to the fabricant’s instructions, the heart rate monitor was set under the participant’s clothes in direct contact with the thorax skin.

### 2.4. Experimental Design

Before the trials, participants had their height and weight measured (in light clothing, without shoes). Later, they performed various lifestyle and simulated working activities. Activities were selected to represent a variety of postures and activity levels between light and moderate, considering some of the activities referred to in the tables from ISO 8996:2004 [10] and practices mentioned in the literature. The designed sequence of activities and their respective classification are detailed in Table 2, and their approximate completion was 75 min.

The variables tested for the model were chosen based on the literature: weight [43,44,45,46,47,48,49,50,51,52,53,54,55], height [47,48,49,52,53,54], age [47,48,49,50,51,52,53,54], sex [47,48,49,50,51,52,53,54,55], fat mass [53,54], fat-free mass [50,53,54], ethnicity [53,56,57], sleep time and quality [17,58,59,60], effort required by the task [53], emotional aspects [61,62], actigraphy counts per minute [43,44,45,46,52,53], personal physical level [63], and heart rate [52,64,65,66]. The variables were screened considering the smallest values of significant effects until obtaining the combination of the variables included in the model.

### 2.5. Data Analysis

Recordings from the devices and information from the questionnaires were gathered and exported to CSV files. Values from cardiac activity, accelerometer counts, and energy expenditure by indirect calorimetry were synchronized and converted to average values per minute.

Model and respective cross-validation analyses were developed using statistical techniques. The modeling process involved variables correlations and establishing a mixed hierarchical model through the software R version 4.0.5 (Ross Ihaka & Robert Gentleman, Auckland, New Zealand). The parameters of variation were estimated using the method cited by Cheung et al. [67], the restricted maximum likelihood (REML).

The adopted parameter of assessment for the model was the bias of standard errors calculated using the difference between model results and standard and literature methods with indirect calorimetry.

#### 2.5.1. Mixed-Effects Regression Modeling

According to Fávero et al. [68], mixed-effects regression models or hierarchical regression models have a primary function in explaining a variable’s behavior over time and identifying its variance components. Hierarchical linear regression is a particular form of a multiple linear regression analysis in which variables are added to the model in sequential steps called “blocks”. This is carried out to statistically “control” the specific variables and see whether adding them improves the model’s ability to predict the criterion variable and investigate a variable’s moderating effect. Accordingly, hierarchical models are multilevel models in which each of these levels is characterized by its submodel, representing the structural relations and residual variability from that level.

Groups characterizing levels of evaluation of the effects of energy expenditure were considered to develop the model. Within lower levels are the activities executed by the participant (considering the movement and activities development), and in a higher level are the participants’ characteristics. During the activities execution, the levels evidence variations in the energy expenditure value. Through the mixed-effect modeling process, it was possible to quantify how much of the variation can be explained by the participant’s intrinsic characteristics (participant’s hierarchical level) and how much comes from the activities’ characteristics by analyzing the activities execution. Furthermore, it was also possible to estimate how much of the energy expenditure variation is caused by other aspects that were not considered in the study.

Variations in energy expenditure, explained by the hierarchical levels, can also result in variations in the mean value, which is known by the intercept of that dependent variable and can result from random variations in the effects of the independent variables. The process for selecting the fixed and random effects from variables was formed by three stages, described below:

(i) After verifying the correlations between the study variables, they were combined and tested into mixed models until reaching a significant correlation. Only random effects were introduced in the intercept. In the following model adjustments, the variable with the least significant effect was excluded, which means that those variables in which the statistical significance test resulted in values close to 0 were eliminated.

This procedure was repeated until obtaining a model in which only significant effects were included (with statistical significance test showing *p*-values < 0.05).

(ii) Variables with square terms and variables interactions were introduced into the model resulted in the first stage. These terms were those with variables excluded in the previous stage.

(iii) In the model from stage 2, random effects from variables that had mixed effects were successively introduced. It was verified which variables corresponded to participants levels and which were from activities levels. The random effects were introduced, then, in the respective level.

(iv) The variable “level of physical activity” was inserted. The model was remade based on steps (i), (ii), and (iii) to verify whether this variable is significant in the model compared to the model without this variable.

The mixed-effect regression modeling process is described in Appendix A.

#### 2.5.2. Cross-Validation and Comparison with Reference Methodologies

The study considered a sample of 3915 movement records in 15 activities being performed by 50 participants. From the records made, it was verified that there were missing data for 175. Thus, these observations were not considered in obtaining and validating the models. The validation procedure was developed considering two phases. The first validation was carried out with an aleatory subsample (around 10%) of the total collected records. The remaining data was used to develop the model.

During a second validation phase, some other methods were chosen to calculate the energy expenditure and compare it with the results from the models. We chose to apply methods from the ISO 8996:2004 [10] and reference regression equations found in the literature [45,69]. Methods from level 3 (evaluation of metabolic rate from heart rate recordings) were selected from the ISO 8996:2004 standard, as they were found to be feasible and applicable for the collected data and protocol. Level 1 methods (classifications according to occupation and the kind of activity) and level 2 (evaluation of metabolic rate using group assessment tables and through the tabulated values for various activities) did not apply to the activity protocol used, and the double-labeled water and direct calorimetry methods (from level 4) were not viable given the resources and time available for this research.

The Swartz et al. (2000) [45] and the Freedson VM3 combination (2011) based on Freedson et al. [43,69] equations were chosen for considering accelerometry counts and being applicable for occupational situations according to the indicated population and scope of activities. The Swartz equation [45] used the wrist and waist accelerometry counts while Freedson [43,69] was applied considering the waist counts. The Freedson VM3 combination (2011) algorithm combines the Freedson VM3 (2011) algorithm [69] with the work–energy algorithm to calculate energy expenditure. The values adopted as a comparison parameter were those obtained from indirect calorimetry.

## 3. Results

The model was developed starting with 15 variables (ethnicity, quality index, anxiety level, physical activity level, subjects high, weight, age, body composition—fat-free and non-fat-free mass, sex, acceleration counts of the wrist, waist, and ankle, and heart rate), and after the removal of nine variables, a model containing only variables with significant effects was obtained.

In a second moment, the quadratic effect of weight was introduced in the selected model (which contained six independent variables). This variable was not in the final model selected but has been widely used in previous research. In fact, the variable turned out to be important when considering that its effect on energy expenditure is quadratic. Subsequently, it was also introduced in the selected model interaction effects of the variable sex to investigate whether part of the effect of the movement variables, of wrist and ankle, was dependent on the sex of the individual. These terms presented significant parameters and were included. After introducing these terms, the parameter associated with fat-free mass proved to be nonsignificant and was removed from the model.

The obtained model can be described by Equation (1) with physical activity level variable, and by Equation (2) without physical activity level variable, and explains the behavior of the energy expenditure according to the included variables: Heart rate (x_1_), Wrist counts per minute (x_2_), Ankle counts per minute (x_3_), Weight’s quadratic term (x_4_²), Physical activity level (x_5_), Female sex: Wrist counts per minute (x_2F1_), Female sex: Ankle counts per minute (x_3F1_). Details on the modeling process can be consulted in the Appendix A.
(1)$$\mathrm{EE}=-3.17+0.000035x_{5}+0.04944x_{1}+0.000047x_{2}+0.000213x_{3}+0.000193x_{4}^{2}-0.000046x_{2F1}-0.000119x_{3F1}+\varepsilon +\pi$$
(2)$$\mathrm{EE}=-2.999+0.04942x_{1}+0.00004736x_{2}+0.0002129x_{3}+0.0001927x_{4}^{2}-0.0000414x_{2F1}-0.0001223x_{3F1}+\varepsilon +\pi$$

### 3.1. Mixed-Effects Regression Model—Estimation and Error

The random effect of the hierarchy level related to movements, considering the explanation capacity, was excluded due to the reduced values. Table 3 illustrates the random effects and the levels analyzed, considering the explanation capacity of the expenditure estimated by the model.

Table 3 presents the results of a model selected in the first stage, which considered only random variations in the intercept. These variations can be interpreted as a general average independent of the effects of the independent variables. It is observed that most of the variation in energy expenditure is caused by aspects related to the participants (31.7%) and the activity performed (32.26%), totaling 63.96% of the explained variation. The research variables do not explain 36.04%, which is related to aspects of momentary execution of the activity, representing the residual variation associated with each specific sample record. The difference of results between the model, with and without the variable “physical activity level”, was negligible, being considered null.

Table 4 presents the fixed effects of the variables estimated using the nonrestricted maximum likelihood method.

Regarding fixed effects, analyzing the intercept, negative values were obtained. Although the intercept is commonly considered an average to which the dependent variable tends when all independent variables assume the value zero, it is possible due to negative values even when they seem inconsistent. In the present study, the negative value of the intercept only indicates a tendency of very low values when the variables tend to zero. This negative value of the energy expenditure must be interpreted with caution. Considering that the basal energy value is around 70% of the individual’s total energy expenditure, the case where the independent variables are settled on zero value cannot be observed. The minimum value of energy expenditure prediction will always correspond to a basal heart rate frequency and the subject’s mass. The intercept should only be seen as a correction of the energy expenditure for the boundaries of the model.

The results presented in Table 4 also verify that the most significant effect among the variables was heart rate (0.0494). Among the movement indicators, it was found that wrist movements had a minor effect than movements measured at the ankle. The finding agrees that in everyday activities, whole-body movements are related to the movements of the legs (measured by ankle counts), and the associated energy costs are higher than those performed by the arms [70].

The results in Table 4 also suggest a difference between male and female subjects in the effect of movements on energy expenditure. Therefore, the movement performed during the activities influences energy expenditure in two separate ways: one depending on the movement counts itself, and the second considering the movement combined with the sex of the subject. Female participants tended to spend approximately 0.000046 less per unit increased in the actigraph record present in the wrist than male participants. For the actigraph located on the ankle, the energy expenditure per unit for female participants tended to be approximately 0.000119 less than those of male subjects. The differences in the body size and muscular strength between genders might explain the variation in intensity and amplitude of the movements [71].

Another result from the model and selecting its terms indicates that linear effects of weight were not significant. However, the variable did show a significant effect when its quadratic form was inserted. Thus, evaluating the two types of effect on energy expenditure suggests that the increase in energy expenditure depends on the subject’s weight. As the coefficient effect is positive, we conclude that energy expenditure increases with weight, but the relationship with the quadratic form means the increase is more expressive for individuals with heavier weights.

In the model selected with random effects only in the intercept, successive random effects from the independent variables were inserted, but the explained variance by these random effects of the variables (heart rate, weight, wrist counts per minute, ankle counts per minute and physical activity level), when introduced in the models described in Equations (1) and (2), are very low. Therefore, it would not be helpful to keep a random term in the model that explains so little of the variance of energy expenditure. Thus, the model that has only random variations in the intercept is more appropriate.

Given this, we have a hierarchical mixed regression model for estimating energy expenditure based on the experiment developed according to Equations (1) and (2). The model was able to estimate the energy expenditure based on the variables collected, pointing out the priorities of these variables by their relevance in the model.

Some sample characteristics did not allow analyzing variables, such as ethnicity, a confounding factor related to energy expenditure, by the literature [57]. Although the sample had seven different nationalities, there was not enough variation to identify such a relationship. The same occurs with age, which is presented in the literature as a significant variable. The analyzed methods use this variable as a factor of relation with the energy expenditure, but the developed models did not show a significant association. Possibly, the sample composition may have prevented an analysis of this variable.

Table 4 presents the contribution of the variables. The physical activity level is the variable with a lower estimated effect and higher *p*-value (0.0000351; *p*-value = 0.0267). Despite a significant association, there is no clear evidence in the literature on how activity level directly influences energy expenditure. For this reason, a version of the model without physical activity was tested. In Table 5, there are no differences in the adjustments and errors, which means that both models’ performances are the same; however, values in Table 4 indicate that the random variation in the intercept increases from 29.86% to 31.70% when removing physical activity from the model, which means that the variable has a relevant influence on the prediction. Differences in the predicting capability are additionally presented in Section 3.2.

Moreover, with the present model having a significant expression on the random term related to the participants, it needs to be readjusted with new data to be enforced to new subjects. The sample model is generalized concerning the variable set and respective structure; however, predicting the results requires some data from the participants. This feature makes the model more accurate and reliable for application in any situation. The adjustment procedure will cause the model to evolve and be perfectly adapted to the new participant group, producing better estimations. This adjustment can be made with a single collected sample, and not a complete protocol or dataset.

### 3.2. Comparison with Standard Techniques and Literature Models

The total sample contained 3740 records and was divided into two sample groups: 3416 records (91.33%) were used to adjust the models, and 324 (8.67%) were used as a validation sample. This last subsample was composed of data from aleatory records referring to 19 participants. For the second validation phase, the correlation coefficient values and standard deviation between each method and indirect calorimetry are presented (Table 6).

The values obtained by the other methods were compared with the values obtained by the calorimetry method through the bias standard error and the standard deviation. These values are represented in Bland–Altman graphs [72] to evaluate the agreement between the estimated values and the values measured by indirect calorimetry, consequently indicating the agreement of each method with the reference method indicated in the standard. 

The Bland–Altman graphs (Figure 2) illustrate the level of agreement between the measured data and the respective methods presented. Figure 2a,b correspond to the hierarchical mixed model with and without physical activity variables. Comparing the model without the variable “level of physical activity” (a) and the model that includes the variable (b), the standard deviation presents no variation, and the Pearson correlation coefficient shows an insignificant change. The dispersion presented by the cloud of values and the bias standard error remains the same. The physical activity being a variable with a significant association, it can be stated that the relevance of this variable requires further studies with larger datasets to be analyzed.

Figure 2c shows the agreement between the measured data and the equation of Swartz et al. (2000) [45]. Finally, Figure 2d corresponds to the concordance results between the measured data and the VM3 (2011) combined equation of Freedson and colleagues [43,69]. Figure 2e illustrates the agreement between the measured data and the data obtained by the heart rate relation method, presented in ISO 8996:2004 [10].

The mixed models (a) and (b) show good agreement with the values of energy expenditure measured by calorimetry because the bias standard error values are closer to zero. Other techniques present a graphically visual distance and the relationship with heart rate (e) method and the (c) Swartz [45] and (d) Freedson VM3 combined [69] equations.

The correlation coefficients shown in Table 6 evidence the association between these methods and indirect calorimetry. The agreement between values is represented by the mean and standard error (bias) proximity to the zero axes. 

Within the standard method (e), the relationship between energy expenditure and heart rate showed the highest bias standard error for the adopted activity protocol, despite being the method with better accuracy among the three standard methods considered for comparison. On the other hand, methods (c) and (d), which are the estimates by the equations, showed higher bias standard errors among all the methods.

The estimates by the Swartz equation [45] showed a general tendency to overestimate the energy expenditure. At the same time, the estimates by the Freedson VM3 (2011) combined equation [69] showed a tendency to underestimate the energy expenditure. 

Therefore, considering all the methods evaluated and respective values of error, dispersion, correlation, and agreement with the indirect calorimetry, the mixed models were revealed to be the most appropriate to estimate the energy expenditure values in this study. The difference between the bias standard error of methods (a) and (b) was not significant, the simplified version (b) without the physical activity level variable being a suitable option to estimate the energy expenditure in the occupational context.

## 4. Discussion 

### 4.1. Energy Expenditure Estimation in the Occupational Context

ISO 8996:2004 [10] is designed to specify different methods for determining metabolic rate in the context of ergonomics of the working environment. Being a reference document, the methods for measuring energy expenditure indicated in ISO 8996:2004 [10] have drawbacks for some instances, especially for field studies. The methods of the observation and screening levels of approach are subject to gross variations between the values in the tables and the actual values. The estimated and tabulated values are appropriate for human averages, being, for men aged 30 years, 70 kg body mass, 1.75 m height, and body surface area 1.80 m²; while for an average of women aged 30 years, 60 kg body mass, 1.70 m height, and body surface area 1.60 m². It is mentioned in the standard [10] that the correction of values is necessary to estimate the metabolic rate for children, the elderly, and people with disabilities, but without specifying or indicating how such corrections should be made.

The errors for using tables are significant, ranging from 5% to 20% error [10]. Such errors are related to environmental conditions, height, weight, gender, and activity characteristics that are being evaluated. Even for a regular working population, individual differences might impose different risk levels when, for example, assessing the heat stress risk using the metabolic rate for the human average [26]. In addition, in ergonomic risk assessment, the evaluation of physical effort and work–rest cycles, the individual adjustments are crucial to prevent overexertion and musculoskeletal-related injuries [28,73].

Furthermore, the standard [10] itself foresees situations where it is necessary to use weighted average values and interpolations when performing resting activities combined with intense activities in an asymmetric distribution. Interpolations are also necessary in cases of applying heart rate values for determining metabolic rate [10]. Heart rate is subject to factors other than physical activity and can easily provide biased results [52,64,65,66]. Another limitation of the standard method, based on heart rate, is the minimum limits of 50 kg and 20 years of age, and maximum limits of 90 kg and 60 years of age, excluding people included in the working population outside this range, i.e., below 50 kg and 20 years of age or above 90 kg and 60 years of age, disregarding, for example, the increase in working age in recent decades. These aspects demand reflections in ergonomics and occupational health in considering a customized way for calculation models to include the variety of activities performed in work situations and the characteristics of the economically active population. The current model presents a significant random effect that comes from the participants. This value increases when we exclude the physical activity level from the model (see Table 3). This reinforces the necessity to adapt the model to the subjects and include variables related to the individuals to improve the accuracy of the predictions.

The model developed reinforced that heart rate is an important variable in determining energy expenditure (Table 4), but contrarily to the standard [10] that includes the dependency on weight and age, it significantly relates to the movement. Despite being one of the variables with a strong correlation with energy expenditure, heart rate has variations that are not easily controlled [14]. Moreover, the linear model proposed by ISO 8996:2004 [10] showed results with errors and dispersion. However, in the models developed, the variable showed paramount relevance in estimation combined with the data regarding movements, as presented in the literature [64,74]. The literature indicates that the limitations of estimation based only on heart rate, especially for light and sedentary activities, can be improved considerably by combining heart rate with actigraphy [75]. It is also the case for activities in which pelvis acceleration is not closely related to whole-body energy expenditure, such as card games and sweeping [64] or free-living activities [74].

There are methods not foreseen in the ISO 8996 standard [10] (such as the use of effort perception scales associated with energy expenditure estimation tables and empirical models) suitable to be enforced in workload assessment that have better performances than the standard methods presented. Some models are developed based on empirical studies of indirect calorimetry and regression techniques involving components of a task or characteristics of a specific group [12,27]. The equations related to movement and activity execution, those by [43,44], widely used in the literature, are some of these examples. The model with uniaxial accelerometer values [44] was used to show that activity patterns can be classified by actigraphy regarding the intensity level of the movements. Nevertheless, the activities monitored were restricted to walking and running on a treadmill, and the sample consisted of 35 persons. The model for triaxial accelerometers [43] was elaborated based on a sample of 50 participants and considered the magnitude vector of the counts as an indicator of the movement quantification. The model proposed by Swartz et al. [45] tried to integrate data from triaxial accelerometers located at the waist and the wrist in a single model, discussing the optimization of the allocation of the devices and the effects of the combined calculations. In our proposal, the modeling process reveals that the counts of the magnitude vectors of the arm and leg actigraphs can better explain variance than the waist counts or the waist combined with the leg counts, which better explain the variance of the global free-living, multitask routines.

Actigraphy is absent from the standards, which indicates an opportunity for it to be explored as an alternative for determining energy expenditure, considering what is already developed and its enforcement potential to occupational contexts. The variables commonly used in equations to estimate energy expenditure are weight, height, age, gender, actigraph counts, body mass index, and heart rate [12,46]. The variables related to the movements, heart rate, and subject’s weight were confirmed to be, under this study’s conditions, those with a more significant influence on energy expenditure estimation. Regarding the sex variable, the natural metabolic differences, body fat and fat-free mass distributions, and the differences in body size must be considered among the differences that explain the association of the variables, as they were in some of the equations analyzed from the literature [76]. The mixed model pointed out possibilities of the influence of sex on the estimation of energy expenditure being related to the differences in the execution of activities between men and women. In Table 4, it is pointed out the negative adjustment relative to sex, considering that women spend less energy than men in the performance of the activities. The model also pointed to a quadratic instead of a direct relationship between subjects’ weight and energy expenditure. It was also important to verify that weight is even more significant than the other related variables, such as fat-free and non-fat-free mass.

One of the advantages of hierarchical mixed modeling is the possibility to analyze the effects of the variables in the model [40,41]. The model reinforced the potential to estimate the energy expenditure based on body movements, which are closely related to the activity performed and the characteristics of each participant. In this sense, another advantage of the model is the possibility of calculation considering these two characteristics—the activity performed and the characteristics of each person. The proposed mixed models, with and without the “physical activity level” variable, change the traditional approaches’ paradigm. It clearly shows that heart rate is an important variable but also includes accelerometer count, their relationship with sex, weight (body plus carried load mass), and physical activity level. Even being the least influential variable in the model, the physical activity level proved significant, indicating a need for further studies on this influence. Either by body characteristics or by metabolic differences [77], among other possible reasons [27,78], it becomes significant in the explanation of the energy expenditure results.

On the other hand, the methods proposed in the expertise approach [79] are more accurate but are neither practical nor very appropriate for field studies, besides being expensive for specific situations. In the case of the double watermarked method, the standard presents only the principle of the method. The minimum recommended execution time for measuring children is six days, for normal adults 12 to 14 days, and for older people, the standard indicates that it may be a more extended period without specifying or estimating it. The standard [10] also admits that the concept is simple in the double-labeled water method, but there are numerous complex details that the user needs to know. However, these details are not mentioned. Considering direct calorimetry, one of the main drawbacks is the difficulty or impossibility of the portability of the equipment for field studies and the high cost [48,79]. Although the improvements of estimation methods increased with technological advances, further studies to explore the causal relationships of energy expenditure variables in the occupational context are more complex than the movement execution analysis. There are intrinsic aspects of the human being and his interaction with the work environment that cannot be easily evaluated [80]. 

Even having a practical and experimental character, this work also has an exploratory aspect since energy expenditure in the occupational context is not trivial. The exploratory aspect triggered the search for contributions from several areas of knowledge crossing with occupational engineering. The current model was specifically developed to be enforced in the occupational context. A simple equation based on variables that are easily measured by any company, though, make the application of the current technique a perfect way to assess the metabolic rate or physical workload. The method only requires heart rate, wrist and ankle counts per minute, weight, and sex. The cross-validation presented high-quality adjustment values (Table 5), and the comparison with other calculation alternatives demonstrated the best accuracy level.

By measuring simple parameters in the field, it will be possible to predict the metabolic rate and improve the assessments based on energy expenditure made in the occupational context. The proposed system for data collection, in the final version, only requires two actigraphs, a heart rate measurement, and basic data. In the occupational field, the applications are enormous, going from managing the workers’ physical exertion, controlling and managing activity levels, preventing exertional heat stress, and other assessments that can be made continuously in real time without interfering with the work activities.

### 4.2. Strengths and Limitations

The possibility of developing the practical component with the available resources was essential to ensure the quality of the data and, consequently, of the developed models. An improvement in this regard would allow evaluating all individuals at the beginning of the day and providing a meal offered and quantified by the research team. Another limitation was the time dedicated to each activity of the protocol, since five minutes is a restricted time for good measurements without compromising the data, even being above the minimum recommendation of three minutes. However, if the time of each activity were longer, the quantity and variety of activities would be reduced. Although the proposed model was developed based on experimental data collected exclusively for this purpose, and has been subjected to data cross-validation, one of the study’s main limitations was the impossibility of testing the proposed models in real context activities and with larger samples.

Nevertheless, the main strength of the research is that the proposed mixed models for determining energy expenditure for occupational settings by actigraphy present results close to the best methods of the reference standard. 

Finally, regarding the model’s applicability, the fact that it should be adjusted for other participants can be reported as both a strength and limitation for its potential applications. The model’s formulation is closed concerning the variables, but to be applied to other samples with high accuracy, it must be previously readjusted. These adjustments can limit its rapid applicability but guarantee its reliable and accurate performance. The proposed model is a model that must evolve to have precision and accuracy. As the term of the aleatory variable depends on the individuals, it will need to be readjusted with the information of other subjects to be applied to them. The model will learn and evolve in each application until it reaches maturity. 

## 5. Conclusions

According to the literature, based on the historical–technological and normative foundation, the use of actigraphy to determine the energy expenditure in a multitasking occupational environment was a real possibility. Experimental research was developed, which allowed replicating some of the methods identified in the literature, enabling data collection to develop estimation models. As a result, an energy expenditure determination model for occupational settings is proposed.

From ISO 8996:2004 [10], we applied estimation methods by the relationship with heart rate and by indirect calorimetry. As a gold standard measurement [76,79], results from the indirect calorimetry were compared to energy expenditure estimation of all other methods to identify the suitable technique in occupational settings. In addition, two of the equations found in the literature were also applied, and the two estimation models were developed by hierarchical mixed regression. The enforcement of Swartz’s (2000) [45] and Freedson’s VM3 combined (2011) [69] equations resulted in overestimation and underestimation, respectively. Even though these equations apply to occupational context, considering multitask assessments, including load carriage, the sample characteristics and the activities used in their development may influence these differences in results, which is one of the limitations of the energy expenditure equations.

The component of the actigraph allocated to the ankle, followed by the wrist, showed the most significant effect on energy expenditure estimation among the actigraphs combination (from the dominant side and waist positions). The difference in the effect of body movements on energy expenditure presented by the hierarchical model among individuals stands out as a relevant factor. Another significant result concerns the effects of body weight identified as nonlinear by a quadratic component in the hierarchical model, which means a relationship where this effect depends on the observed value of the variable. For the case of this model, the energy expenditure tends to increase with the weight, the variations being more expressive for individuals with higher weights. This relationship needs to be studied in greater depth.

The activity level variable showed a relationship with the energy expenditure. Depending on how active the individual is, it will influence the energy expenditure and, consequently, his/her physical performance at work regarding the execution of movements. As the literature points to an indirect relation, and as the contribution for the energy expenditure is minor, the version of the model without the physical activity level is the final proposal to be enforced in the occupational context. Finally, the two developed models present themselves as a possible way to study people’s energy expenditure in an occupational context, making the development of the system based on actigraphy feasible for this purpose, with a comparable error concerning the methods presented in the ISO 8996:2014 standard [10].

The hierarchical mixed model showed 30% of the variation in the expenditure related to the participants’ factors, 33% to the performance of the activity, and 37% as aspects not explored in the study. Therefore, the 63% (30% + 33%) effect explanation capacity can be considered good, but the other 37% of unexplored aspects are opportunities for future studies. Further research is required to deepen studies about the physical activity level effects on energy expenditure, increase the sample to study the association of age, race, and quadratic impact of weight, and explore the sex variable effect on movements.

## Figures and Tables

**Figure 1 ijerph-18-10419-f001:**
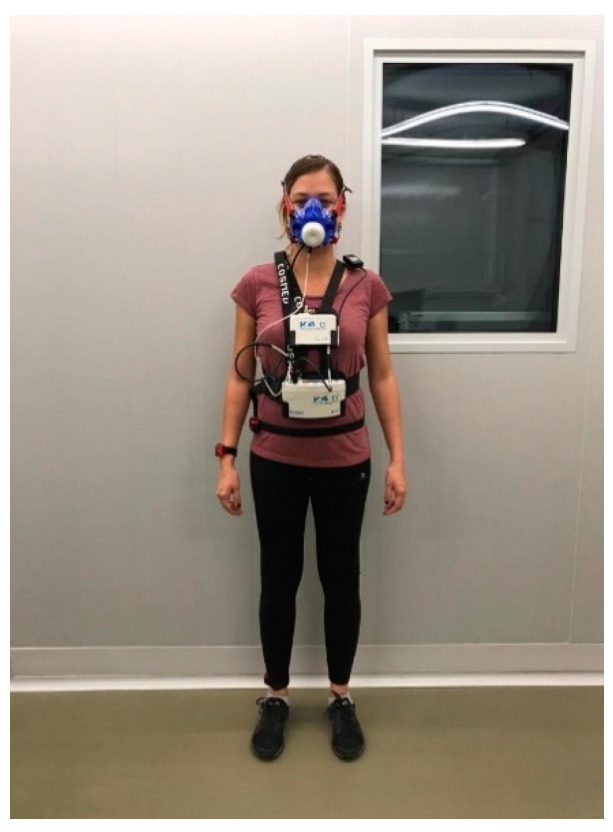
Equipment disposition during the experiments.

**Figure 2 ijerph-18-10419-f002:**
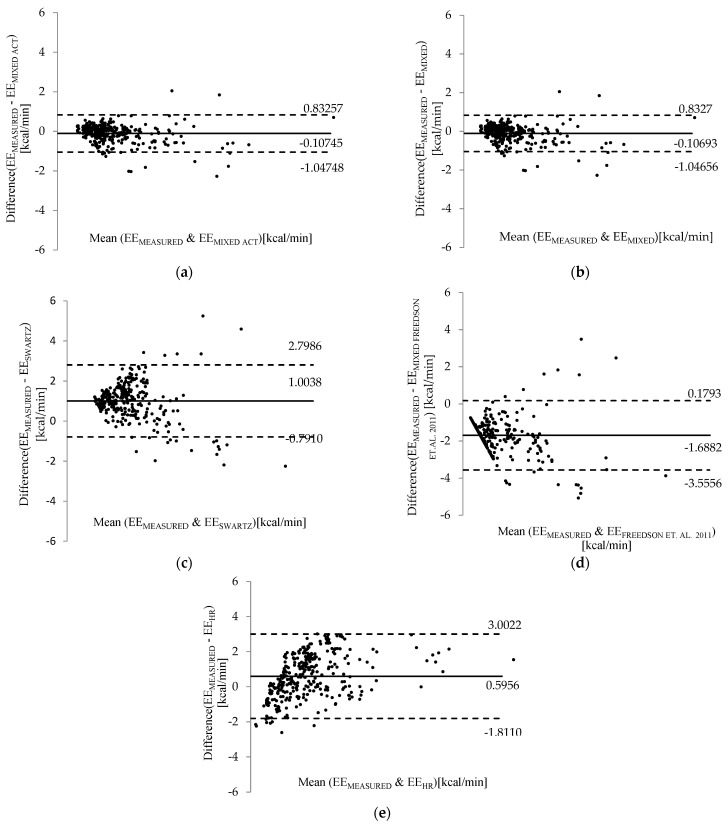
Bland–Altman plot of agreement between measured data and (**a**) the hierarchical mixed model with activity variable, (**b**) the hierarchical mixed model without physical activity level variable, (**c**) the Swartz equation (2000) (**d**) Freedson’s VM3 combined equation (2011), and (**e**) the heart rate relation method ISO 8996:2004.

**Table 1 ijerph-18-10419-t001:** Participants characteristics.

Variables	Total (*n* = 50)		Males (*n* = 25)		Females (*n* = 25)	
	Mean	SD	Mean	SD	Mean	SD
Age (years)	29.84	5.12	30.40	5.74	29.28	4.46
Height (cm)	170.20	9.54	176.16	8.32	164.24	6.55
Weight (kg)	69.29	14.06	77.98	13.15	60.60	8.58
FFM (kg)	52.34	12.65	61.68	10.71	43.00	5.51
FM (kg)	16.54	5.59	15.96	6.31	17.12	4.84

Note: SD—standard deviation; FFM—fat-free mass; FM—fat mass.

**Table 2 ijerph-18-10419-t002:** Detailed description of protocol of activities.

Activity Sequence	Description	Duration (min)	Type of Activity
1	Lying	10	Basal
2	Sitting, doing computer work	5	Basal
3	Standing, playing with cards	5	Multitask
4	Standing, moving up and down a 2 kg load, metronome: 40 bits/min	5	Multitask
5	Sitting, watching a video	5	Basal
6	Sweeping	5	Multitask
7	Sitting–standing 10 times	Free	Multitask
8	Sitting, watching a video	5	Basal
9	Moving plastic boxes with a 5 kg load	5	Multitask
10	Moving plastic boxes with a 10 kg load	5	Multitask
11	Sitting, watching a video	5	Basal
12	Slow walking (from the lab to the stairs)	Free	Displacement
13	Stairs (go down and up four floors)	Free	Displacement
14	Slow walking (from the stairs to the lab)	Free	Displacement
15	Sitting	5	Basal

**Table 3 ijerph-18-10419-t003:** Random effects of investigated levels on the intercept.

Level	Parameters	Including Physical Activity Level		Excluding Physical Activity Level	
		Explained Variance	Relative Values	Explained Variance	Relative Values
Participants	Intercept	0.2439	29.86%	0.2660	31.70%
Activities	Intercept	0.2706	33.12%	0.2707	32.26%
Residual	-	0.3024	37.01%	0.3024	36.04%
	Total	0.8169	100.00%	0.8391	100.00%

**Table 4 ijerph-18-10419-t004:** Fixed effects of the variables before the adjustments.

Variable	Including Physical Activity Level		Excluding Physical Activity Level	
	Estimated Effect	*p*-Value	Estimated Effect	*p*-Value
Intercept	−3.17	<0.001	−2.999	<0.001
Heart rate (x_1_)	0.04944	<0.001	0.04942	<0.001
Wrist counts per minute (x_2_)	0.0000473	<0.001	0.00004736	<0.001
Ankle counts per minute (x_3_)	0.000213	<0.001	0.0002129	<0.001
Weight’s quadratic term (x_4_²)	0.000193	<0.001	0.0001927	<0.001
Physical activity level (x_5_)	0.0000351	0.0267	−	−
Female sex: Wrist counts per minute (x_2__F__1_)	−0.000046	<0.001	−0.0000458	<0.001
Female sex: Ankle counts per minute (x_3__F__1_)	−0.000119	<0.001	−0.0001191	<0.001

**Table 5 ijerph-18-10419-t005:** Measures of adjustment and error of the models.

Measure	Excluding Physical Activity Level Variable	Including Physical Activity Level Variable
R^2^	0.8325592	0.8324836
Bias	−0.0329973	−0.03372094
MAE	0.4397435	0.4397387
RMSE	0.613014	0.6131523
Standard Deviation	0.6130721	0.6131713

**Table 6 ijerph-18-10419-t006:** Mean of the errors of the total expenditure of the cycle between the methods applied in relation to the energy expenditure by indirect calorimetry.

Energy Expenditure Assessment Method	Bias Standard Error [kcal]	Pearson Correlation Coefficient (r)	Standard Deviation [kcal]
(a) Hierarchical mixed-regression model excluding “physical activity level”	−0.0330	0.9129	0.6131
(b) Hierarchical mixed-regression model including “physical activity level”	−0.03372	0.9128	0.6132
(c) Swartz equation	1.0038	0.7948	0.9157
(d) Freedson VM3 combination equation	−1.6882	0.7779	0.9528
(e) Heart rate estimation	0.5956	0.7812	1.2278

## Data Availability

The data presented in this study are available on request from the corresponding author.

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
