# Peer review of "Modeling Energy Expenditure Estimation in Occupational Context by Actigraphy: A Multi Regression Mixed-Effects Model"

_ijerph, 2021, doi:10.3390/ijerph181910419_

Round 1

Reviewer 1 Report

Dear authors,

your manuscript faces an interesting topic related to prediction of energy requirements for healthy individuals in an occupational context by means of actigraphy.

However, there are some aspect, just in my opinion, that should be taken into account before considering for publication.

First comments:

  • Avoiding to refer to the models as mathematical equations or mathematical regression. I mean that in the introduction you could substitute <<…several mathematical equations…>> by <<several models (or methods)>>, as well as <<developed by mathematical regression>> by <<developed by regression>> (twice in the introduction). We know that there is mathematic under the models so it is not necessary to highlight this fact.
  • In the last paragraph of the introduction you introduce your main goal, as it has to be, but I would erase the beginning <<As a result>> and I would begin with <<This work aims at…>>. Here you also write that you perform a cross-validation with respect to de gold standard methodologies. In my opinion the use of gold standard term in this context is not appropriate since in medicine and health sciences a gold standard is the absolute truth and you are only comparing with other well founded methodologies.
  • Regarding to the participants, why only 50? You also write that they were volunteers but, were they volunteers within a random sample or only accessible people?
  • At the end of the subsection 2.3 you write that the variables for the model were chosen according to the literature. Please, a reference citation would be pertinent.
  • Regarding to subsections 2.4.1 and 2.4.2, I am not sure if they should be comprised in subsection 2.4 but with not a so detailed description. In my opinion this information is not relevant for clinicians or physicians (non statisticians) and could be boring. In fact, since you add an appendix at the end of the manuscript, this description should be there as it seems to be. Furthermore, in my opinion, the appendix should not be part of the manuscript but a supplementary material for volunteer access.
  • Table 4 indicates p-values lower than 2.10-6, but it is usual when you obtain very low p-values to indicate p<0.001.
  • With respect to the Appendix A, please avoid the dots within equation (9). Use sub index to write β0 and others in text as you do in the equations. Here you also have to review the equations citation because, for instance, you write <<equations (10) to (13) were combined into equation (1) from which equation (10) was obtained>> when I understand that they were combined in equation (9) to obtain equation (18). Please review. You also refer that the random variations u10, u20 and u01 were considered equal to 0, u01 is not defined never (I think you are talking about u00).

There are also several concerns related to the model description.

  • First, the notation for the explanatory variables is confuse for the weight’s quadratic term (x4²) even for Female sex wrist and ankle counts per minute x2:F1 and x3:F1 because in the appendix A you use another notation and you refer to gender not only to female as I think it is. This is very confuse because you are estimating a linear model but with this notation it seems another thing.
  • Then, once you justify that there is no evidence of normality problems despite that presence biases of homoscedasticity I don’t understand why you perform the same regression models for the logarithmic transformation if you critique and seem to discard these models. Furthermore in this situation is the centred logarithmic transformation the more used one and the interpretation of the parameter, once you undone the transformation it used to be consistent with the original ones.
  • I also would erase equations (1) to (8) since they don’t throw more relevant information that it is already included in Tables 4 and 5.
  • Anyway if you decide to maintain them in the manuscript you should do some corrections. I mean: equation (3) has a double log in the right term when it should be only one, if it were correct, because Equations (3) and (4) are confuse since they seem to indicate that you are applying logarithms to equations (1) and (2) when you are really estimating the linear model for the logarithm transformation. It is weird because you write that equations (5) and (6) are obtained from these ones and it is not true with this formulation. Finally, despite equations (7) and (8) seems to be right from (5) and (6), an additive form of these ones (additive exponential) is more usual and appropriate.

Finally, I think that you have an adequate structure for the manuscript but you are not exploding properly all the sections. For instance, your result section seems not to depict all the numerical results that you refer in the discussion and even in the conclusions. Please review and make consistent your results section with respect to the discussion and conclusions ones.

Author Response

The authors are grateful for the reviewer’s effort in reading and critically analysing the manuscript. The observations were certainly valuable for its improvement. We have carefully considered all the suggestions and provide a point-to-point response to each of them.

Comment 

Response 

Section 

First comments: 

Avoiding to refer to the models as mathematical equations or mathematical regression. I mean that in the introduction you could substitute <<…several mathematical equations…>> by <<several models (or methods)>>, as well as <<developed by mathematical regression>> by <<developed by regression>> (twice in the introduction). We know that there is mathematic under the models so it is not necessary to highlight this fact. 

We thank for the suggestion. Accordingly, we eliminated the term “mathematical” throughout the introduction section.  

Introduction 

In the last paragraph of the introduction you introduce your main goal, as it has to be, but I would erase the beginning <<As a result>> and I would begin with <<This work aims at…>>.  

It was revised accordingly. 

Introduction  

Here you also write that you perform a cross-validation with respect to de gold standard methodologies. In my opinion the use of gold standard term in this context is not appropriate since in medicine and health sciences a gold standard is the absolute truth and you are only comparing with other well founded methodologies. 

The phrase was modified as follows: “This work aims to propose an actigraphy-based model for energy expenditure estimation in the occupational context based on hierarchical regression modelling and cross-validate the results obtained with the gold standard (indirect calorimetry) and other validated methodologies (normative guidelines from ISO 8996:2004 and predictive equations).” 

Introduction

Regarding to the participants, why only 50? You also write that they were volunteers but, were they volunteers within a random sample or only accessible people? 

Regarding the sample, it was admitted that the size is a limitation of the study, since there were some limitations concerning the availability of the equipment, and the access to the subjects.

However, the collected data per participant is large since we have records per minute, in a protocol of 14 activities performed by more than 60 minutes.  

To clarify the selection process, we included a new subsection 2.1 Sampling, explaining the process selection and respective exclusion criteria. 

Regarding the model development, we revised the text to emphasise the fact that the model should be adjusted for other participants. Its formulation is closed concerning the variables, but to be applied to other subjects with accuracy it must be previously readjusted.  

Materials and Methods 

At the end of the subsection 2.3 you write that the variables for the model were chosen according to the literature. Please, a reference citation would be pertinent. 

The references 45 to 68 support the inclusion of the variables, and were respectively placed in the text next to each of them:

“weight [45-57], height [49-51,54-56], age [49-56], sex [49-57], fat mass [55,56], free fat mass [52,55,56], ethnicity [55,58,59], sleep time and quality [17,60-62], effort required by the task [55], emotional aspects [63,64], actigraphy counts per minute [45-48,54,55], personal physical level [65], and heart rate [54,66-68].”  

Materials and Methods 

Regarding to subsections 2.4.1 and 2.4.2, I am not sure if they should be comprised in subsection 2.4 but with not a so detailed description. In my opinion this information is not relevant for clinicians or physicians (non-statisticians) and could be boring. In fact, since you add an appendix at the end of the manuscript, this description should be there as it seems to be. Furthermore, in my opinion, the appendix should not be part of the manuscript but a supplementary material for volunteer access. 

We believe that the explanation of the modelling process is important to understand the contribution of the variables, respective interaction and decision process for adopting the final model.

We found available literature laconic concerning the developed modelling processes. Thus, we intend to overcome these limitations and try to point out the importance of guiding future users on how to use the model and adjust the equations. 

Materials and Methods / Appendix 

Table 4 indicates p-values lower than 2.10-6, but it is usual when you obtain very low p-values to indicate p<0.001. 

We agree with the observation, the values in Table 4 were changed accordingly. 

 Results 

With respect to the Appendix A, please avoid the dots within equation (9). Use sub index to write β0 and others in text as you do in the equations. Here you also have to review the equations citation because, for instance, you write <<equations (10) to (13) were combined into equation (1) from which equation (10) was obtained>> when I understand that they were combined in equation (9) to obtain equation (18). Please review. You also refer that the random variations u10, u20 and u01 were considered equal to 0, u01 is not defined never (I think you are talking about u00). 

The dots from equation (9) were excluded as suggested. Similarly, subindexes were corrected, and equation numbers were revised.  

Concerning the random variations, they were also revised and corrected, u01 does not exist, it was a typing error; u10, u20 and u00 were maintained. 

Specifically, u00 is not equal to 0 in this stage and is a random term that the model does not explain. However, it is maintained in the model.  

 Appendix 

There are also several concerns related to the model description. 

First, the notation for the explanatory variables is confuse for the weight’s quadratic term (x4²) even for Female sex wrist and ankle counts per minute x2:F1 and x3:F1 because in the appendix A you use another notation and you refer to gender not only to female as I think it is. This is very confuse because you are estimating a linear model but with this notation it seems another thing. 

We thank the observations. The notation can indeed be ambiguous, implying that there is a division by using the two dots ":". We tried to make it more explicit by using the subscript form of the F symbol and uniformising the notation of this variable. The variable did not intend to call the gender variability but include the female difference on it. 

 Appendix 

Then, once you justify that there is no evidence of normality problems despite that presence biases of homoscedasticity I don’t understand why you perform the same regression models for the logarithmic transformation if you critique and seem to discard these models. Furthermore, in this situation is the centered logarithmic transformation the more used one and the interpretation of the parameter, once you undone the transformation it used to be consistent with the original ones. 

We are not sure if we clearly understood the pointed problem. However, the model remains robust despite of some eventual normality issues, as supported by the references [74,75]. With all, as presented, the transformation was done to prevent any issue, but the model remains consistent and is the same. 

 Results 

I also would erase equations (1) to (8) since they don’t throw more relevant information that it is already included in Tables 4 and 5. 

We appreciate and thank the suggestion. However, once again we believe that the detail in the modelling process is an important feature in the manuscript. It provides detailed explanation about the included variables and their contribution.

In addition, it presents the final process of adjustment which is very important to the people who want to reproduce and apply the model.

However, after discussing this point with the research team, we opted to locate most of the equations to the Appendix section and keep equations (1) and (2). 

 Results 

Anyway if you decide to maintain them in the manuscript you should do some corrections. I mean: equation (3) has a double log in the right term when it should be only one, if it were correct, because Equations (3) and (4) are confuse since they seem to indicate that you are applying logarithms to equations (1) and (2) when you are really estimating the linear model for the logarithm transformation. It is weird because you write that equations (5) and (6) are obtained from these ones and it is not true with this formulation. Finally, despite equations (7) and (8) seems to be right from (5) and (6), an additive form of these ones (additive exponential) is more usual and appropriate. 

It was decided to maintain the equations (1) and (2) and the rest of relevant equations in the Appendix section. Accordingly, related text and respective sections were revised.  

 Results 

Finally, I think that you have an adequate structure for the manuscript but you are not exploding properly all the sections. For instance, your result section seems not to depict all the numerical results that you refer in the discussion and even in the conclusions. Please review and make consistent your results section with respect to the discussion and conclusions ones. 

Concerning this aspect, we deeply reviewed the content of the model and decided to include more information to better support this content: 

1) data from the model validation and respective interpretation were added to the results section. We believe that this section makes clear the quality of the model even before its testing. A better explanation was provided concerning the method and the data that were used to test and validate it. 

2) interpretation of the modelling process and a more detailed explanation of the enforcement was included in the discussion and conclusion sections. 

Results/ Discussion/ Conclusions 

Reviewer 2 Report

Summary

The authors presented the results of an econometric study (a multi regression mixed-effect model) of factors influencing energy expenditure by young and healthy people. The most important thing was the work-related energy expenditure. Based on a literature review on: 1 / analytical / statistical tools; 2 / equipment used to study energy expenditure, an own study was constructed, the results of which, according to the authors, are more important than those presented in the literature.

1 / The manuscript’s strengths:

- / thorough literature review;

- / properly selected statistical tool for calculations;

- / selection of other tools for measuring energy expenditure;

- / selecting the most important (?) factors influencing energy expenditure;

- / describing the study in a very detailed way - step by step;

- / comparison of the obtained results with the results obtained by other researchers;

- / the study is very reliably designed;

-/ the obtained results are important as they numerically confirm the expectation of a relationship between daily activity and energy expenditure. Some of the results confirm numerically the differences resulting from anatomy and physiology. 

2 / The manuscript's weaknesses:

- / the research sample is limited to young people around the age of 30 - this is probably too small a range for the work-related energy expenditure survey. I have the impression that the results for people in older age groups would be more important;

- / in the introduction, there was no evidence to show, based on the literature, how the own results differ from those obtained in other studies;

- / the study has been described in a very detailed way, however, it is a purely "technical" description with no interpretation;

- / it has not been fully demonstrated that the results obtained by the authors are more important than the results obtained with other methods and tools described in the literature;

- / Were the factors, and which ones, influencing energy expenditure, which turned out to be insignificant, checked?

- / the description of the results is laconic - this is especially important in the case of comparisons with the results obtained in other studies, which were designed differently in various respects. Is it not possible to provide an interpretation in the comparisons of the individual measurement methods?

- / in the discussion and in the conclusions, the interpretation was presented in a laconic way - the possibility of using the obtained results was not presented. Moreover, it has not been shown what the obtained results have for ergonomics, economics of occupational health, and public health;

- / the text is rather aimed at researchers using statistical tools, and therefore difficult to understand for decision-makers who would like to use the obtained results in practice.

- / on page 11 there is fig. 37c in the description - there is no such numbering in the drawings.

Author Response

The authors are grateful for the reviewer’s effort in reading and critically analysing the manuscript. The observations were valuable for its improvement. We have carefully considered all the suggestions and provide a point-to-point response to each of them.

Comments 

Responses 

Section 

Summary 

The authors presented the results of an econometric study (a multi regression mixed-effect model) of factors influencing energy expenditure by young and healthy people. The most important thing was the work-related energy expenditure. Based on a literature review on 1 / analytical / statistical tools; 2 / equipment used to study energy expenditure, an own study was constructed, the results of which, according to the authors, are more important than those presented in the literature. 

1 / The manuscript’s strengths: 

- / thorough literature review; 

 - / properly selected statistical tool for calculations; 

- / selection of other tools for measuring energy expenditure; 

- / selecting the most important (?) factors influencing energy expenditure; 

- / describing the study in a very detailed way - step by step; 

- / comparison of the obtained results with the results obtained by other researchers;- / the study is very reliably designed; 

-/ the obtained results are important as they numerically confirm the expectation of a relationship between daily activity and energy expenditure. Some of the results confirm numerically the differences resulting from anatomy and physiology.  

We thank the reviewer for the positive comments on the manuscript. 

2 / The manuscript's weaknesses: 

- / the research sample is limited to young people around the age of 30 - this is probably too small a range for the work-related energy expenditure survey. I have the impression that the results for people in older age groups would be more important; 

We defined the sample characteristics but did not stratify according to age. With all, the sample was tested in a huge range of activities with different levels of physical exertion, and to create the model, this was the most important feature. 

To make it more accurate, it is necessary to use the aleatory independent term, which explains around 36% of the variance. This means that the independent characteristics of the subjects are crucial to getting higher accuracy, so our approach suggests the need to include the information of the subject before being applied. This strategy brings accuracy but requires some extra work before applying the model to other samples. It can be seen as a weakness and a strength, but it was the solution to have a simplified model. 

 The proposed model is a model that must evolve to have precision and accuracy. As the term of the aleatory variable depends on the individuals, it will be needed to readjust with the information of other subjects to be applied to them. The model will learn and evolve in each application until it reaches maturity. 

Materials and Methods 

- / in the introduction, there was no evidence to show, based on the literature, how the own results differ from those obtained in other studies; 

It was included an extended version of the model validation that uses an 8% sample of the data collected (not included in the modelling process) to validate the model.

The model comparison was also done considering the differences between the gold standard measure of indirect calorimetry and the model predictions concerning the same 8% of the data. 

Introduction/ Results 

- / the study has been described in a very detailed way; however, it is a purely "technical" description with no interpretation; 

We appreciate the observations. The text from the Material and methods and Results sections, were extensively revised in order to improve the interpretation. 

- / it has not been fully demonstrated that the results obtained by the authors are more important than the results obtained with other methods and tools described in the literature; 

We tried to include more information from other studies, which was difficult due to the differences in study design.  However, the research compared different techniques, and it was shown that the model performs best compared with these other methodologies. The most appropriate solution, which evidence is scarce in literature as well, is a practical verification comparing the different models against a reference method (indirect calorimetry).  

Discussion 

- / Were the factors, and which ones, influencing energy expenditure, which turned out to be insignificant, checked? 

In the initial modelling process, some variables were removed because they did not explain the results. The stages leading to the selection of the variables to be included are outlined at the beginning of the Results section.

We tried to clarify the process indicating which variables were removed from all the tested and combined variables. 

 Results 

- / the description of the results is laconic - this is especially important in the case of comparisons with the results obtained in other studies, which were designed differently in various respects. Is it not possible to provide an interpretation in the comparisons of the individual measurement methods? 

Indeed, the designs of the other studies are very different, so we believe we should focus on the practical application of different models compared.  

Results  

- / in the discussion and in the conclusions, the interpretation was presented in a laconic way - the possibility of using the obtained results was not presented. Moreover, it has not been shown what the obtained results have for ergonomics, economics of occupational health, and public health; 

The interpretation was respectively clarified in the discussion and conclusions sections.

As the results are more accurate than the other methods, the enforcement in OSH brings the possibility to improve the assessments based on metabolic rate measurements or estimations.

This model brings new possibilities of assessment in the field of OSH and any other that requires an accurate and simplified assessment of energy expenditure in daily routines (e.g., Medicine, Nutrition, Sports, Physiology).  However, to be applied within Public Health, the model needs to evolve into a form that considers the individual variation.  

Discussion  

- / the text is rather aimed at researchers using statistical tools, and therefore difficult to understand for decision-makers who would like to use the obtained results in practice. 

The model brings a contribution to the field in several aspects, concerning the explanation of the variables, exploring an alternative technique to predict energy expenditure and so on, but to be enforced at the current stage, it will require some expertise. The model is closed; however, the adjustments need to be done for future applications. The decision making will need to see the contributions as the understanding of the tested variables and their relevance. 

Discussion /Conclusions  

- / on page 11 there is fig. 37c in the description - there is no such numbering in the drawings. 

Reference to fig. 37 was corrected accordingly.  

Results  
